Disturbance has lasting effects on functional traits and diversity of grassland plant communities

Smith Ellen A. 1
Holden Emily M. 1 eholden@ualberta.ca
Brown Charlotte 1 2
Cahill Jr James F. 1
1 Department of Biological Sciences, University of Alberta , Edmonton, Alberta , Canada
2 Desert Laboratory on Tumamoc Hill, University of Arizona , Tucson, Arizona , United States
Jaffé Rodolfo
Electronic publication date: 2022 Mar 25
Publication date: 2022
Volume: 10
Electronic Location ID: e13179
Received 2021 Oct 5; Accepted 2022 Mar 7
Copyright: © 2022 Smith et al.
Copyright year: 2022
Copyright holder: Smith et al.
License: This is an open access article distributed under the terms of the Creative Commons Attribution License, which permits unrestricted use, distribution, reproduction and adaptation in any medium and for any purpose provided that it is properly attributed. For attribution, the original author(s), title, publication source (PeerJ) and either DOI or URL of the article must be cited.
License URL: https://creativecommons.org/licenses/by/4.0/

Keywords: Functional diversity, Functional traits, Successional communities, Disturbance, Grasslands, Plant strategies

Funding: NSERC Discovery NSERC Discovery Accelerator ACA Grants in Biodiversity University of Alberta NSERC CGS‐M NSERC PGS‐D This work was supported by a NSERC Discovery Grant (James F. Cahill Jr), NSERC Discovery Accelerator Supplement grant (James F. Cahill Jr), two ACA Grants in Biodiversity (Emily M. Holden and James F. Cahill Jr/Charlotte Brown and James F. Cahill Jr), University of Alberta Master’s scholarship (CB), NSERC CGS‐M scholarship (Charlotte Brown), and a NSERC PGS‐D scholarship (Charlotte Brown). The funders had no role in study design, data collection and analysis, decision to publish, or preparation of the manuscript.

==============================
Background

Localized disturbances within grasslands alter biological properties and may shift species composition. For example, rare species in established communities may become dominant in successional communities if they exhibit traits well-suited to disturbance conditions. Although the idea that plant species exhibit different trait ‘strategies’ is well established, it is unclear how ecological selection for specific traits may change as a function of disturbance. Further, there is little data available testing whether disturbances select for single trait-characters within communities (homogenization), or allow multiple trait-types to persist (diversification). We investigated how (a) traits and (b) functional diversity of post-disturbance gap communities compared to those in adjacent undisturbed grasslands, and (c) if altered functional diversity resulted in the homogenization or diversification of functional traits.

Methods

Here we emulate the impacts of an extreme disturbance in a native grassland site. We measured plant community composition of twelve paired 50 × 50 cm plots (24 total) in Alberta, Canada. Each pair consisted of one undisturbed plot and one which had all plants terminated 2 years prior. We used species abundances and a local trait database to calculate community weighted means for maximum height, specific leaf area, specific root length, leaf nitrogen percent, and root nitrogen percent. To test the impacts of disturbance on community functional traits, we calculated functional diversity measures and compared them between disturbed and undisturbed communities.

Results

Within 2 years, species richness and evenness in disturbed communities had recovered and was equivalent to undisturbed communities. However, disturbed and undisturbed communities had distinct community compositions, resulting in lower functional divergence in disturbed plots. Further, disturbance was linked to increases in community-weighted mean trait values for resource-acquisitive traits, such as specific leaf area, and leaf and root nitrogen.

Discussion

Disturbance had lasting effects on the functional traits and diversity of communities, despite traditional biodiversity measures such as richness and evenness recovering within 2 years. The trait space of gap communities shifted compared to undisturbed communities such that gap communities were dominated by traits enhancing resource uptake and growth rates. Overall, these results show that short-term disturbance fundamentally changes the functional character of early-successional communities, even if they superficially appear recovered.

Introduction

Habitat heterogeneity in grasslands is maintained in part by small-scale mortality inducing disturbances, such as fossorial mammal activity (Davidson & Lightfoot, 2008; Davidson, Detling & Brown, 2012), drought (Godfree et al., 2011), overgrazing (di Virgilio & Morales, 2016), pathogens (Mursinoff & Tack, 2017), and even herbicide drift (Fried, Villers & Porcher, 2018). In extreme instances, disturbance can result in the total mortality of mature vegetation (e.g. Cahill, 2003). Though each of these disturbances have unique characteristics, they all result in the localized death of established vegetation, creating vegetation “gaps” (Suding, 2001). Gaps result in changes to abiotic soil conditions and competitive hierarchies (Suding & Goldberg, 2001; Suding, 2001), leading to changes in species diversity and community composition.

Examining traits can shed light on the specific impacts of disturbance on plant communities. Traits are measurable individual morphological, physiological, or phenological characteristics that provide insight into life history strategies and influence plant ranges and species interactions (Voille et al., 2007; Cadotte, Carscadden & Mirotchnick, 2011). “Functional traits” specifically impact growth, reproduction and survival (Voille et al., 2007). At small scales, trait differences can influence local diversity patterns by determining competitive outcomes (Liu et al., 2015). Traits can also provide insight as to species resource acquisition strategies by indicating species location on the fast-slow plant economics spectrum, where “fast” strategies prioritize resource acquisition and growth while “slow” strategies favour stress tolerance and longevity (Reich, 2014). Measuring functional traits after a disturbance (such as those which create gaps) can reveal how a community has changed in functional strategies and can be used to examine the resulting community’s functional diversity. Cadotte (2007) describes gap species as prioritizing rapid resource uptake, whereas non-gap species take a slower, tolerance-based approach. For example, specific leaf area (SLA) values are likely to be larger in post-disturbance successional communities as high SLA is correlated to fast growth and low competitive ability (Kunstler et al., 2016). Additionally, Loughnan & Gilbert (2017) found that SLA may be involved with shifts in competitive ability as greater SLA is associated with more sunlight acquisition and thus increased energy acquisition. Thus, changes in communities after mortality are likely due to differences in the trait profiles of successional species.

It is important to note that these changes in species composition and community-level traits changes may not be reflected in the total number of species in the community (i.e. species richness). Disturbance may change the nature of community assembly in successional communities by altering the process by which species from the regional pool are able to colonize local communities (HilleRisLambers et al., 2012; Escobedo et al., 2021). Disturbance changes the environmental filters which constrain species dispersal and recruitment to communities (Myers et al., 2015; Brown & Cahill, 2020); in this case we would expect the resulting community to contain species that possess functional traits better suited to the new set of environmental filters, changing the community-level traits and functional diversity of the disturbed community. Thus, if taxonomic measures like species richness or evenness remain unchanged after disturbance, they may mask functional differences and result in a mirage of stability when, in fact, the community has undergone great change. This emphasizes the importance of studying functional traits and quantifying functional diversity to understand community drivers.

Functional diversity is a subset of biological diversity which measures the traits present in an ecosystem (Tilman, 2001). It is distinct from taxonomic diversity as it reflects the breadth of the functional space occupied by species in a community (Rosenfeld, 2002; Villéger, Mason & Mouillot, 2008), whereas taxonomic diversity indicates the number of species in a given community separate from any measure of community function (Laliberté & Legendre, 2010). Some functional diversity measures are not greatly influenced by taxonomic measures such as species richness (Laliberté & Legendre, 2010) and can offer a more nuanced approach to understanding community characteristics. Disturbance affects functional diversity (Parreira de Castro, Dolédec & Callisto, 2018) and can potentially result in functional homogenization, or selection for similar trait-characters in species colonizing in disturbed communities (Olden et al., 2004). However, resource fluxes associated with disturbance (Davis, Grime & Thompson, 2000) may lead to trait diversification within colonizing species to allow for more efficient utilization of the increased resources available in disturbed communities (Chapman, Childers & Vallino, 2016; Jentsch & White, 2019). As functional diversity is associated with the provisioning of ecosystem services (Díaz et al., 2007; Roscher et al., 2012; Pakeman, 2014), the homogenizing or diversifying effects of disturbance on functional diversity must be better understood.

If species in post-disturbance gap areas are more likely to possess traits which support faster resource acquisition (e.g. Cadotte, 2007; Kunstler et al., 2016), then shifts in trait distributions after disturbance may alter functional diversity by changing the success of different trait suites to favor “fast” traits prioritizing resource acquisition at the local scale. However, it is unclear which traits differ between plants in disturbed and undisturbed areas, and how these different strategies are reflected in functional diversity. To resolve these outstanding questions, we ask: How do functional traits of a grassland community change after disturbance?

How does functional diversity change after disturbance?

If functional distributions do change, will it result in the homogenization or diversification of community-level traits?

If disturbance causes shifts in habitat filters, then we expect to observe shifts in the functional trait space of the resulting community. If disturbance constrains the competitive strategies that succeed in successional communities (for instance, favouring plants with resource-acquisitive traits), we would expect changes in trait values (i.e. community-weighted means) and potentially the homogenization of community-level traits. Alternatively, if species turnover after disturbance drives shifts in community traits, we then expect to observe both taxonomic and functional trait shifts, which could in turn diversify community-level traits.

Methods

Study site

Our study site was located in the Roy Berg Kinsella Research Ranch in Kinsella, AB, Canada (53°5′N, 111°33′W). Data collection took place in a native grassland dominated by Hesperostipa curtiseta (Hitchc.) Barkworth, Festuca hallii (Vasey) Piper and Poa pratensis (L.) (Brown, Oppon & Cahill, 2019). The site is part of the Aspen Parkland ecoregion, a savanna-type habitat characterized by a mosaic of mixed-grass prairie and trembling aspen (Populus tremuloides Michx.). The field site is periodically grazed by cattle with a heavy grazing event in October 2019 and a light grazing event occurring in May 2020.

Study design

We sampled 24 plots (50 cm × 50 cm), arranged in 12 blocks of paired plots, which were originally established in 2016 (Brown & Cahill, 2020). Each block contained an undisturbed plot, where no standing vegetation was terminated, and a disturbed plot, where all standing vegetation was terminated. Disturbed plots were created in May 2016, by having all biomass trimmed to the soil surface and an undiluted glyphosate herbicide (Roundup©, Bayer, Leverkusen, Germany) liberally applied to the remaining stems to ensure death of the resident vegetation. This treatment was maintained through August 2018 with any regenerating plant materials or newly germinated recruits trimmed and painted with herbicide during the growing seasons. This treatment does not represent any specific natural event, and instead is testing the extreme event of complete removal of the resident vegetation, without a soil disturbance. Additionally, this treatment does not remove the seed bank present at the site (Brown & Cahill, 2020) and thus does not affect new plant recruitment from seed after the treatment window (see Grubb, 1977).

Plant community composition was measured in July 2018 by visually estimating percent cover of all species present within each plot. Two years post-treatment, in July 2020, plot pairs were revisited, by which time vegetation had regrown in the disturbed plots. Communities were colonized through enhanced germination of seed from the existing seed bank and seed dispersal from surrounding areas (Brown & Cahill, 2020), Community composition was measured using percent cover estimation of each species. Sedges were unable to be identified to species and were recorded as “Carex spp.”.

Plant traits

Disturbance can alter abiotic conditions (Suding & Goldberg, 2001); however, here we focus on its impacts on vegetative traits. Plant trait data came from a database developed principally at this field site (Cahill, 2020), thus representing local trait data. Details of trait measures are found in Cahill (2020), but largely follow the methods outlined in Cornelissen et al. (2003) and Pérez-Harguindeguy et al. (2013). Here we focused on five traits which encompass aspects of plant structure and above- and below ground resource acquisition: maximum height, specific leaf area (SLA), specific root length (SRL), leaf nitrogen percentage (N %), and root N %. See Cahill (2020) for trait definitions. These traits were chosen to provide a holistic scope of above and below ground functional strategies with reference to trait groups described by Cadotte (2017). Site-specific trait data was available in the database for all species except for Cirsium vulgare (Savi) Tenore, Collomia linearis Nuttall, Gentianella amarella (Linnaeus) Börner, Sonchus arvensis Linnaeus, and two unidentified forbs. Trait data was obtained for species that represent at least 93% of total composition, which exceeds the 80% threshold standard for trait studies (Pakeman & Quested, 2007).

Trait profiles and functional diversity

To characterize communities by their traits, we first calculated community weighted means (CWMs) for five traits: SLA, maximum height, leaf N %, root N %, and SRL. CWMs are the average value of a given trait in a community weighted by the abundance of all species possessing said trait (Lavorel et al., 2008), and are useful for understanding community properties and dynamics, as well as quantifying community change (Garnier et al., 2004, 2007; Louault et al., 2005). CWMs were calculated at the species-level. While disturbance is likely to result in intraspecific variability in functional traits, here we focus on species-level values as a first-level test to detect if disturbance does result in persistent change to community functional traits. We encourage future studies to collect trait data from individual plants across species to quantify how intraspecific trait variation changes with disturbance regimes. Outlier trait values (values more than three standard deviations away from the species’ mean) were removed prior to calculating CWM. We then quantified the functional diversity of communities using functional richness (FRic), functional evenness (FEve), functional divergence (FDiv), and Rao’s quadratic entropy (Q) (Villéger, Mason & Mouillot, 2008; Mouchet et al., 2010; Table 1). These metrics allow us to characterize the volume (via FRic), evenness (via FEve), and spread (via FDiv and Q) of the communities’ functional traits in multidimensional space. By using these multivariate descriptors of communities, we are able to compare communities’ character before and after disturbance, as well as to determine if community traits become more or less homogenous as a result of disturbance. We used package vegan (Oksanen et al., 2017) in program R (v 4.0.0; R Development Core Team, 2020) to compute species richness and evenness, using methods from Oksanen (2020). We used package FD (Laliberté, Legendre & Shipley, 2014) to compute CWMs. We also computed FRic, FDiv, FEve, and Rao’s Q using package FD.

Table 1 Functional diversity metric definitions.

Functional diversity metric definitions	
Functional diversity (FD)	The functional space occupied by species, where axes are functional features (Rosenfeld, 2002; Villéger, Mason & Mouillot, 2008). The functional differences between a group of species (Tilman, 2001)	
Functional richness (FRic)	The breadth of functional space filled by a communities (Villéger, Mason & Mouillot, 2008)	
Functional evenness (FEve)	The evenness of the distribution of abundances and functional features of species (Villéger, Mason & Mouillot, 2008)	
Functional divergence (FDiv)	The average distance of species abundances from the centre of functional space (Mouchet et al., 2010)	
Rao’s quadratic entropy (Q)	The average functional distance between two randomly selected species in a group (Mouchet et al., 2010).	

Statistical analysis

To determine if community traits differed between control and disturbed treatments after recovery, we used five separate linear mixed models (LMMs). The CWM’s for SLA, height, leaf N %, root N %, and SRL were used as response variables, plot type (disturbed or undisturbed) was a fixed effect and block (i.e., disturbed and undisturbed plot location) was a random effect. “Block” was included as a random effect in models to account for spatial autocorrelation in community data as disturbed and undisturbed plots were adjacent. All LMMs were run using an underlying normal distribution with the lme4 (Bates et al., 2015) and lmerTest (Kuznetsova, Brockhoff & Christensen, 2017) packages. We also used four separate LMMs to quantify how functional diversity measures, namely FRic, FDiv, FEve, and Rao’s Q, differed between disturbance types. FRic, FDiv, FEve, and Roa’s Q were used as response variables, block was included as a random effect and plot type (disturbed or undisturbed) was a fixed effect in all models. Functional richness was log-transformed to fit assumptions of normality. To determine whether disturbed and undisturbed communities had different community compositions, we conducted a permutational multivariate analysis of variance (PERMANOVA), with plot as a strata, using packages vegan and RVAideMemoire (Herve, 2021).

Results

Between 2018 and 2020, total species richness in undisturbed plots decreased from 55 to 52 while average species richness in disturbed plots increased from 0 in 2018 to 53 in 2020 (Table S1). Six species were found only in disturbed communities (Androsace septentrionalis, Cirsium arvense, Mulgedium pulchellum, Thalictrum venulosum, an unidentified Brassicaceae plant, and an unidentified herbaceous dicot; Table S1). Androsace septentrionalis, Cirsium arvense, and Mulgedium pulchellum are commonly found in disturbed areas (Tannas, 2004). In particular, Cirsium arvense is associated high rates of reproduction and dispersal. A total of 15 species were found only in undisturbed communities (Table S1). These species, except Fallopia convolvulus, are all native to Alberta (Desmet & Brouilet, 2013).

Despite persistent effects on local diversity, and although disturbed plots began with zero species present in 2018, within 2 years there was no significant difference between plot-level (alpha) richness and evenness between disturbed and undisturbed plots (richness = 17, evenness = 0.82; Fig. 1). However, underlying this similarity are differences in species composition (R2 = 0.24, F = 7.06, p = 0.001; Fig. 2), with one third of plant species unique to either treatment type (Fig. 1; Table S1). Gap communities were segregated from undisturbed communities in multivariate space, with the exception of one disturbed/undisturbed pair (pair #12) (Fig. 2).

Figure 1 Average species evenness and richness of disturbed and undisturbed plots in 2018 and 2020.

In disturbed plots, all biomass was trimmed to the soil surface and a glyphosate herbicide (Roundup©) applied to the remaining stems to ensure death of the resident vegetation from 2016–2018. Plots were allowed to regenerate from 2018–2020. Undisturbed plots had no standing vegetation removed. Species evenness and richness was recorded in the summer of 2020. Error bars represent standard error.

Figure 2 Metric multidimensional scaling (MDS) of disturbed and undisturbed community composition.

Data points denote 2020 plant community composition in disturbed and undisturbed plots. In disturbed plots, all biomass was trimmed to the soil surface and a glyphosate herbicide (Roundup©) applied to the remaining stems to ensure death of the resident vegetation from 2016–2018. Plots were allowed to regenerate from 2018–2020. Undisturbed plots had no standing vegetation removed. Community composition was assessed in the summer of 2020.

Community weighted means of SLA, leaf N %, and root N % differed between disturbed and undisturbed plots, with disturbed plots typically having trait values more consistent with the ‘fast’ end of the leaf economics spectrum. Specifically, disturbed plots had significantly larger CWM for SLA, leaf N %, root N %, and there was a trend towards larger root SRL (Fig. 3; Table 2). There was no difference in CWM of maximum height, indicating a more rapid recovery to overall physiognomy relative to other functional characteristics of the communities.

Figure 3 Mean trait values in 2020 disturbed and undisturbed communities.

Comparisons of specific root length, maximum height, specific leaf area, root nitrogen, and leaf nitrogen between disturbed and undisturbed communities. Error bars represent standard error. Associated statistics are found in Table 2.

Table 2 Results of the linear mixed models (LMM) for traits.

	Disturbed plot mean ± SE	Undisturbed plot mean ± SE	df	p-value	
Maximum height (cm)	35.78 (±1.05)	36.30 (±1.16)	11	0.66	
Specific leaf area (cm2/g)	139.96 (±2.99)	119.34 (±2.99)	11	2.64 × 10 –5	
Specific root length (cm/g)	5,612.82 (±448.19)	4,550.30 (±570.50)	11	0.09	
Leaf nitrogen (%)	1.92 (±0.04)	1.74 (±0.06)	11	0.01	
Root nitrogen (%)	1.05 (±0.03)	0.95 (±0.04)	11	0.02	
Note:

Community weighted means for specific leaf area (SLA), height, shoot percent nitrogen (shoot N %), root percent nitrogen (root N %), and specific root length (SRL) were used as response variables in separate models. Plot pair (i.e., disturbed and undisturbed plot location) was a random effect and plot type (disturbed or undisturbed) was used as the fixed effect. SE stands for standard error and df denotes degrees of freedom. Bold type indicates significant results.

Gap communities were functionally distinct from undisturbed communities after 2 years of recovery. Undisturbed communities showed higher FDiv values (FDiv = 0.08 ± 0.02, p = 0.0020; Fig. 4). FEve (FEve = 0.07 ± 0.02, p = 0.0129; Fig. 4), FRic (difference = 1.07 ± 1.16, p = 0.376; Fig. 4), and Rao’s Q (difference = 0.004 ± 0.002, p = 0.0733; Fig. 4) were not significantly different among the two treatments (Table 3).

Figure 4 Estimates of functional diversity values in 2020 disturbed and undisturbed communities.

Comparisons of functional richness, functional evenness, functional divergence, and Rao’s quadratic entropy between disturbed and undisturbed communities. Functional richness was log-transformed to meet the assumption of normality and was multiplied by negative one. Error bars represent standard error. Associated statistics are found in Table 3.

Table 3 Results of the linear mixed models (LMM) for functional diversity.

	Disturbed plot mean ± SE	Undisturbed plot mean ± SE	df	p-value	
FRic (log transformed)	–39.96 (±0.92)	–38.75 (±1.22)	11	0.345	
FEve	0.66 (±0.02)	0.69 (±0.02)	11	0.218	
FDiv	0.73 (±0.02)	0.81 (±0.02)	11	0.003	
Rao’s Q	0.02 (±0.001)	0.02 (±0.002)	11	0.078	
Note:

Functional richness (FRic), functional divergence (FDiv), functional evenness (FEve), and Rao’s Q were used as response variables in separate models. Plot pair (i.e., disturbed and undisturbed plot location) was a random effect and plot type (disturbed or undisturbed) was a fixed effect. SE stands for standard error and df denotes degrees of freedom. Bold type indicates significant results.

Discussion

Although local communities recovered in species richness after only 2 years, there were a number of legacies of disturbance found in functional traits. There are persistent effects of disturbance on local diversity, suggesting successional communities are shaped by a combination of local dispersal and niche conditions. However, given competition can be reduced by disturbance (Wilson & Tilman, 1993), dispersal most likely limits recruitment in gap communities. At the same time, at the plot level we find there is great stability in species richness and evenness, suggesting the local structure of communities in this system is highly stable but the species who fill each “role” are variable. Thus, the functional differences we observe are not due to fundamental changes in the dominance structure (i.e. evenness) of successional communities, but rather due to species filtering.

Traits after disturbance

Consistent with a priori expectations, we show disturbance leads to functional shifts favouring species with higher SLA, leaf N %, and root N %, which are important for the rapid acquisition of resources. SLA and leaf nitrogen content are widely viewed as indicative of a fast-growing, rapid nutrient acquisition strategy (Wright et al., 2004; Liu et al., 2017). Greater leaf nitrogen content works in concert with high SLA to increase energy exploitation through improved photosynthetic capacity, as high leaf N content is necessary for photosynthetic protein functioning (Wright et al., 2004). High root N is also indicative of quick resource acquisition as it is related to high root respiration, high foraging ability, and low root longevity (Craine et al., 2002; Reich, 2014; Roumet et al., 2016; McCormack et al., 2017). SRL, while higher in gap communities, was not significantly different between plot types. Typically SRL indicates potential resource uptake per root mass investment (Reich, 2014), and high SRL is associated with quick growth, high foraging capacity, and lower root longevity (Comas & Eissenstat, 2004; Roumet et al., 2016); as such, that gap communities did not show significantly higher SRL is contrary to our expectations. Overall, the higher levels of SLA, leaf N %, and root N % among the disturbed plots signal post-disturbance conditions permit the success of quick-growing plants that can quickly access resources. Thus, it is likely that the spatial variability of disturbances in this system are an important means to create functional variation across the landscape.

Height variation in grasslands is a significant predictor of species richness and community productivity (Brown & Cahill, 2019), yet we found that maximum height recovers quickly (within 2 years) after disturbance and is not significantly different among successional and undisturbed communities. This is surprising as greater height is thought to be indicative of greater competitive ability (Givnish, 1995; Cornelissen et al., 2003; Falster & Westoby, 2003).

Functional diversity after disturbance

Disturbance causes a number of changes to functional diversity, with functional diversity typically increasing in gap communities (Purschke et al., 2013; Eler et al., 2018). However, we observed that disturbance either decreased or did not affect functional diversity measures. There was no difference in FRic values between the disturbed and undisturbed plots, indicating both treatments filled the same amount of functional niche space (Schleuter et al., 2010). Similarly, FEve did not differ between treatments, denoting that species abundances and/or the functional distances between species were equally even in disturbed and undisturbed plots. However, FDiv values were significantly higher in undisturbed communities. Lower FDiv values in the disturbed plots signify that the most abundant species in disturbed plots had more homogenous trait ranges. Thus, while disturbance does not affect the breadth or evenness of community functional traits, it does lower FDiv, promoting homogenization of functional diversity.

Conclusions

Overall, we found while traditional metrics of community composition such as species richness and evenness recovered within 2 years of disturbance, there were persistent impacts of disturbance on community-level traits and functional diversity. SLA, leaf N %, and root N % values were significantly larger in the disturbed condition, supporting the conclusion that species in gap areas are more likely to possess traits that support faster resource acquisition. The prevalence of individuals possessing these “fast” traits was reflected in distinct differences in community membership among undisturbed and gap communities. Disturbance also impacted functional diversity by promoting homogenization of community’s functional traits. In all, this work suggests functional trait shifts from small disturbances are a critical mechanism for maintaining spatial heterogeneity in grassland systems, even as species richness and evenness recover.

Supplemental Information

Supplemental Information 1 Species present in 2018 and 2020 plots.

Click here for additional data file.

Supplemental Information 2 Comparison of linear mixed model and linear model outputs for community weighted mean traits in disturbed and undisturbed plots.

Community weighted means for specific leaf area (SLA), height, shoot percent nitrogen (shoot N %), root percent nitrogen (root N %), and specific root length (SRL) were used as response variables in separate models. Plot pair (i.e., disturbed and undisturbed plot location) was included as a random effect in linear mixed models. Plot type (disturbed or undisturbed) was the fixed effect in linear and linear mixed models. SE stands for standard error. Bold type indicates significant results.

Click here for additional data file.

Supplemental Information 3 Comparison of linear mixed model and linear model outputs for community weighted mean traits in disturbed and undisturbed plots.

Functional richness (FRic), functional divergence (FDiv), functional evenness (FEve), and Rao’s Q were used as response variables in separate models. Plot pair (i.e., disturbed and undisturbed plot location) was included as a random effect in linear mixed models. Plot type (disturbed or undisturbed) was the fixed effect in linear and linear mixed models. SE stands for standard error. Bold type indicates significant results.

Click here for additional data file.

Supplemental Information 4 Species accumulation curve for 2020 data.

Species richness data from disturbed and undisturbed plots was used to create the curve.

Click here for additional data file.

We would like to thank K. Hardman, T. Blenkinsopp, I. Peetoom Heida, and T. Barber-Cross for their assistance in the field. We would like to thank S. Sugden for his assistance with the data analysis and the selection of functional traits.

Additional Information and Declarations

Competing Interests

Author Contributions

Data Availability

The authors declare that they have no competing interests.

Ellen A. Smith conceived and designed the experiments, performed the experiments, analyzed the data, prepared figures and/or tables, authored or reviewed drafts of the paper, and approved the final draft.

Emily M. Holden conceived and designed the experiments, performed the experiments, analyzed the data, prepared figures and/or tables, authored or reviewed drafts of the paper, and approved the final draft.

Charlotte Brown performed the experiments, analyzed the data, authored or reviewed drafts of the paper, and approved the final draft.

James F. Cahill Jr conceived and designed the experiments, authored or reviewed drafts of the paper, and approved the final draft.

The following information was supplied regarding data availability:

The data is available at GitHub: https://github.com/emilmhold/Trait-profiles-and-functional-diversity-following-disturbance-in-a-mixed-grassland.

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
