# Peer review of "Disturbance has lasting effects on functional traits and diversity of grassland plant communities"

_PeerJ, doi:10.7717/peerj.13179_

## Round 0.1 · original submission · Major Revisions

Please carefully address the comments raised by both reviewers, paying special attention to those mentioning your experimental design.

·

Basic reporting

The article is written clearly, has adequate structure and a clear rationale. Literature references are adequate, but the number of figures can be reduced by joining figures in a single plate. Furthermore, figures may be improved by adding p-values (actually this information is placed in tables 2 and 3, and this alteration enables the exclusion of the mentioned tables)

Experimental design

Major flaws in experimental design must be addressed in future versions:
1- No data about abiotic changes in disturbed/undisturbed plots (soils, microclimate, ..) are presented, although the introduction highlights the context between abiotic changes due to gaps/disturbances and its consequences on species composition (e.g., l. 63, l. 77)

2- What about intraspecific trait variation in response to disturbed/undisturbed plots? Plant species differ regarding functional traits, but phenotypic variation along environmental gradients (disturbances) are common phenomens and may affect CWMs of functional traits in disturbed and undisturbed plots.

3- Modelling: 12 CWM traits values or functional diversity values are modelled in function of disturbance regimes using plot pair as a random effect - may overfitting be an issue here?

4- small plot size (50 x 50 cm): I am not familiar with the specific vegetation analyzed in this study, but
I wonder, if the plot size is sufficient to sample diversity of this habitat type. Some statistics on that (species-area relationships etc) would be highly welcome to show if minimum area for phytosociological surveys is sampled.

Validity of the findings

The study is really interesting, but small sample area (3m2 only) and comments on statistics (previous section) make me wonder about the validity of the presented findings. At minimum, supporting statistics should be presented as outlined above.

Additional comments

No comments.

Reviewer 2 ·

Basic reporting

Smith et al report the findings from an interesting study comparing the trait-space occupied by established and post-disturbance grassland vegetation. The findings are novel and I look forward to seeing this in it's final form. My biggest concern is the language surrounding the disturbance treatments, I typically think of grassland disturbance as one that removes/alters above ground cover, but does not result in terminating the plants. This was not the case, I didn't realize that the vegetation was completely terminated in the treatment plots until I was into the methods. This doesn't necessarily take away from the novelty of the study, but it does need to be clear that this is a study comparing established vegetation with what is found two-years into the recovery/recolonization from a bare soil condition- which is very different from a grazing or fire type disturbance that I would typically associate with grasslands. To me this is a matter of the temporal age of the stands being compared - tracking the recovery of similarly aged plants is different than comparing established and recolonized communities.

Specific comments:
L38 - "vegetation removed" to "plants terminated"- big difference here. I would interpret the original wording to mean that the plants were still alive and they tracked the regrowth. Instead all plants were terminated and they tracked recolonization

L46 - functional repeats
L53 - Again taking issue with the disturbance wording - the functional character of recolonized vs established communities is different.
L152-3. Wouldn't successive herbicide applications also terminate plants germinating from the seed back at that time? Yes the seed bank and seed rain is still there, but anything colonizing it during that two year treatment window was terminated?
L155- change to post-treatment
L179- curious about the outliers that were removed
L185-187 - There is no before/after disturbance data presented. There is a comparison between recolonized and established communities
L196 - What distributions were used for the LMM's for each response variable?
L234 - There is no test of a legacy of disturbance here. Again - recolonized vs established communities
L282 - I'd hardly say this is a study of lasting impacts - two years of recovery is still very early in the successional trajectories of grasslands
L289- Couldn't agree more - but change to functional trait shifts resulting from different successional stages - or something to that effect
Tables - Fix functional richness definition

Experimental design

Solid aside from previous comments re: wording surrounding the treatments and interpretation

Validity of the findings

See above

Additional comments

See above

---

## Round 0.2 · Major Revisions

As you will see the Reviewer is not happy with your revision. I agree with him, and would need to see his comments addressed in full to be able to make a final decision. I do not understand why you included plot location as a random effect instead of treating them as replicates. Also,
I checked your R code and found a few issues, including wrong model names (GLMM instead of LMM) and no data being loaded at the beginning of each script. I strongly suggest the authors create a GitHub (or any other open-access) repository containing all R scripts and datasets carefully organized. Each script should run independently of other scripts, unless sourced appropriately.

·

Basic reporting

No additional comments.

Experimental design

During previous round, I mentioned four concerns related to methodology/experimental design:
1- no data about environmental changes in disturbed plots: authors revised manuscript accordingly
2- intraspecific trait variation was not considered: authors revised accordingly, stating that the study as a 'first-level test', but failed on adding comments regarding follow-up research
3- overfitting, as 12 observations were modelled by a fixed and a random effect: authors do not provide additional tests
4- small sample area: Authors do not provide species accumulation curves or species-area relationship to address my concerns. In their response, authors cite two studies that attest appropriate plot size - one is about soil seed banks (and not community composition), for the other one, reference data are lacking.

Validity of the findings

As outlined during the previous round, I am not convinced about the validity of the presented findings, and, as detailed, little efforts have been done to improve the manuscript.

Additional comments

No comments.

---

## Round 0.3 · Minor Revisions

I thank you for the detail revision of your manuscript, which I believe has been substantially improved. I only a have a few additional minor requests:

- Please address the comments made by Reviewer 2.

- Please shorten your abstract and mention there some of the issues raised by the Reviewer (like your sample sizes and your ability to survey your plant communities).

- Also I suggest changing your title with one reflecting the main finding. Something like: "Disturbance has lasting effects on functional traits and diversity of temperate grassland plant communities". This is just an idea, please feel free to use a different one.

·

Basic reporting

OK

Experimental design

Ok

Validity of the findings

OK

Additional comments

Sorry for delay, but I was in holiday!

Reviewer 2 ·

Basic reporting

I recall commenting on this before and it still stands – disturbance is used way too often and way too loosely throughout the manuscript. It’s a good study, but use of disturbance and variations on this word in this way hides the true nature of the experimental design and results in an overreach. In many places it would be much more accurate to use the “gap” terminology – i.e. L116 – e.g. change to species colonizing gaps. Additionally, I find the manuscript overall lacking in essential explanations regarding how these gap areas were revegetated – was this colonization by seed, vegetative spread from edges, or even recruitment from plants that weren’t effectively terminated (which could very well happened).
Some specific comments:
L 53 – Over reach on claim that all disturbances would result in this same effect. This is one type of a disturbance that effectively assesses functional trends in early succession. The language needs to be changed
L58 – such as insert as
L84 -85 – Recovery after mortality would be better here
Study design – It would be good to explicitly articulate that it’s a split plot design
L147 – Give the manufacturer info and application concentrations for the Roundup
L151 – add “without a soil disturbance” to the extreme event clause
L151-153 – So how did you handle the seedling that did emerge during the termination window? I’m assuming that any newly germinated individuals were also terminated – but that could be explicitly stated here – if so, it did affect the seed bank because recruits were terminated. Theres a window of potential recruitment here that needs to be addressed in how it was handled and what effect it would have on the study outcomes
L156 – The word regenerated is used here – that means that there was growth from established plants, which it says above that they were all terminated. Make it clear that this was growth that resulted from new seedling recruitment – if not and the treatment didn’t kill all of the plants or that there’s evidence of tillering from outside of the plots, that needs to be clear. Great line where calling them “gap” plots would make a whole lot more sense

Experimental design

See comments above - especially re: explicitly stating that it was a split-plot design

Validity of the findings

see comments above

Additional comments

see comments above

---

## Round 0.4 · accepted · Accept

I`m happy to accept your manuscript. Thank you for the last revisions.